# A Mixed Methods Study Describing the Quality of Healthcare Received by Transgender and Gender Nonconforming Patients at a Large Integrated Health System

**DOI:** 10.3390/healthcare9050530

**Published:** 2021-05-02

**Authors:** Suma Vupputuri, Stacie L. Daugherty, Kalvin Yu, Alphonse J. Derus, Laura E. Vasquez, Ayanna Wells, Christine Truong, E. W. Emanuel

**Affiliations:** 1Kaiser Permanente Mid-Atlantic States, Mid-Atlantic Permanente Medical Group, Rockville, MD 20852, USA; alphonse.derus@kp.org (A.J.D.); laura.vasquez85@gmail.com (L.E.V.); a.l.wells1222@gmail.com (A.W.); christinextruong@gmail.com (C.T.); ew.emanuel@kp.org (E.W.E.); 2Division of Cardiology, Department of Medicine, University of Colorado School of Medicine, Aurora, CO 80045, USA; stacie.daugherty@ucdenver.edu; 3Kaiser Permanente Southern California, Southern California Permanente Medical Group, Pasadena, CA 91101, USA; kalvin.c.yu@gmail.com

**Keywords:** gender identity, transgender, gender nonconforming, health system, electronic medical record, focus group, quality, and healthcare

## Abstract

Transgender and gender nonconforming (TGNC) patients have been seeking medical care in higher numbers and have faced unique social, personal, and health issues that affect the quality of care they receive. The purpose of this study was to conduct a mixed-methods study to describe TGNC care at Kaiser Permanente Mid-Atlantic States, a large integrated health system. We used a transgender registry to describe a TGNC patient population and compared healthcare utilization between TGNC patients and non-TGNC patients. Four focus groups were also conducted among 28 patients. Atlas.ti software was used to code and analyze themes for the qualitative analysis. Among the 282 adults TGNC patients, the mean age was 32.6 years. Of the study sample, 59% were White, and 27% were Black. TGNC patients demonstrated an increased use of email/telephone visits and the online patient portal and more cancellations and no-shows compared to non-TGNC controls. Of the 28 TGNC patients who participated in the focus groups, 39% identified as female, 21% as a transman, and 18% as non-binary/genderqueer. Participants were predominantly White (68%), highly educated (74%), and reported use of hormones (89%). Themes that emerged from our qualitative analysis included: limited availability of TGNC information; positive and negative sentiments regarding patient–provider interactions; issues with case management; limited access to care; lack of coordination of care; negative staff experiences. We identified specific areas in a health system to improve the quality of care of TGNC patients, including specific TGNC training for providers and staff, a source of TGNC information/resources, and hiring and training TGNC-specific case managers.

## 1. Introduction

The process of gender transition to align a person’s physical sex with their gender identity can vary widely. Some TGNC persons may seek medical transition, while others may wish to transition only socially without medical intervention. The former comprises a heterogeneous group of healthcare consumers who may seek therapies, such as hormone treatment, facial hair removal, speech modification therapy, surgery, or some combination of these procedures.

Recent evidence from a meta-regression of population-based probability samples [1] showed a dramatic yearly increase in the number of transgender adults in the US between 2007 and 2016. However, the increase in the number of TGNC persons is not believed to be an actual increase but a result of the societal shift of increased visibility and awareness, resulting in more people openly identifying as TGNC [1]. The pooled estimate of TGNC persons in the US from data across years (which is likely underestimated because of the increasing prevalence in recent years) is reported at 0.4% or approximately 1.3 million Americans. This conservative estimate is likely to represent younger adults, and future nationally representative surveys will likely identify higher numbers of TGNC individuals [1].

Accordingly, TGNC patients are seeking medical care in greater numbers, potentially owing to greater visibility and acceptance as well as changes in health insurance policies/coverage. To stay in step with these healthcare utilization trends, health systems need to mainstream TGNC care [2]. The quality of care received by this vulnerable patient population remains understudied.

There are many issues unique to TGNC persons that affect the quality of healthcare they receive. Due to a lack of infrastructure support and little formal medical education of culturally competent services, TGNC persons experience barriers in accessing quality healthcare. For example, because their preferred name and gender identity may be different from their legal name and gender, issues of clinic staff using inappropriate name and pronouns could lead to stigmatization, low self-esteem, and negative effects on a patient’s overall mental health [3]. This could have serious implications on transgender patients seeking care when they need it. Most notably, discrimination in the form of disrespect, harassment, violence, or even the refusal of health services frequently prevents transgender persons from seeking healthcare. There have not been any studies that document patients’ perspectives on healthcare quality within a health system where patients have equal access to healthcare.

The purpose of this study was to describe the insured TGNC population at a large integrated health system using electronic medical record (EMR) data and to assess barriers and facilitators to quality transgender healthcare from the patient perspective using patient focus groups. The overarching goal is to provide feedback for health systems developing TGNC health initiatives on what challenges to prioritize to provide quality healthcare to the TGNC patient population.

## 2. Materials and Methods

### 2.1. Design

This study employed a mixed-methods design with two distinct components—quantitative and qualitative—to describe and explore characteristics and issues relating to TGNC healthcare.

### 2.2. Quantitative Component

The quantitative component of the study included patients aged ≥18 years, identified by the Epic (EMR platform) Transgender Registry at KP Mid-Atlantic States (KPMAS) on 15 October 2017. This was a clinical registry designed to help treat and manage patients currently enrolled in the health system. The registry identified patients based on if: (1) they were alive; (2) were currently enrollment at KP; and had either (3a) an ICD diagnosis indicating transgender status (ICD9:302.85, 302.50, 302.6, 302.3, 302.53, 302.51, 302.52; and ICD10: F64.1, F64.2, F64.8, F64.9, Z87.890); or (3b) a history of transgender surgery/procedure (internal KP code). While we did not conduct a validation of the registry’s ascertainment of transgender patients, at the time of the study, all patients in the registry were known to the Transgender Health Services Program Director and Coordinator (co-authors E.W.E. and A.W.) to be transgender or gender nonconforming. Data used for the quantitative component of the analysis was exempt from informed consent.

We conducted a cross-sectional, descriptive analysis of transgender patients. These variables were assessed during a two-year lookback from the date the registry was accessed. We assessed data on demographics, service area, enrollment and healthcare utilization, and transgender-specific characteristics, such as the number of transgender diagnoses and procedures and hormone therapy. These variables were assessed during a two-year lookback from the date the registry was accessed. For context in interpreting descriptive characteristics, we conducted a 1:10 frequency match of patients in the TGNC cohort to non-TGNC patients by age (within one year) and non-missing race/ethnicity. However, since there has not been any validation done of the TGNC registry or its algorithm, we chose not to conduct statistical tests or attempt to interpret significant differences based on this preliminary data.

### 2.3. Qualitative Component

For the qualitative component of the study, the focus group interview guide was developed based on a comprehensive literature review of previous qualitative studies to elicit relevant information from participants. Broad themes included questions relating to their experiences with KPMAS healthcare, experiences with transition, healthcare barriers/facilitators, expectations, and satisfaction. Focus groups and scripts were reviewed and approved by the Director and Program Manager of the Transgender Health Program at KPMAS.

We invited 254 adult patients (identified in the KP Transgender Registry) to participate in focus groups in April 2017 via the online patient portal. A repeated message was sent out in June 2017. A total of 56 patients responded by email or phone call, and 39 patients agreed to participate in 1 of 4 focus groups. Of these, 28 patients attended the focus group discussions and completed a brief survey on demographic and treatment characteristics. (Focus group patient participants were not linked to EMR data). All participants signed an informed consent form agreeing to participate in focus group discussions. Participants received a $40 incentive for their time. The protocol (for both the quantitative and qualitative components of the study) was reviewed and approved by the KPMAS institutional review board.

Focus group discussions were approximately 2–3 h in duration, with 6–8 patients in each. The focus groups were originally planned for 2 h in duration. However, in every discussion group, additional time was required to allow participants to continue conversations, share additional experiences, and ask other participants questions. According to the interview guide, a moderator posed questions with adjustments to the order of topics and wording to facilitate the discussion. Non-verbal messages and the dynamics of each group discussion were observed and recorded. Sessions were audio-recorded and transcribed.

Modified grounded theory [4] was used to conduct inductive analyses using ATLAS-ti. Two independent coders reviewed the transcripts twice to establish emergent concepts and develop a list of unique codes. Three coders then used this list of codes to review and independently code each of the four focus group transcripts. The coded transcripts were merged and cleaned before analysis. Percent agreement and Krippendorff’s Cu-α were calculated to estimate agreement and disagreement, respectively, across coders and to determine intercoder reliability.

## 3. Results

### 3.1. Quantitative Component: Descriptive Analysis of the TGNC Registry Population

The descriptive cross-sectional analysis of patients in the “Transgender Registry” at the time of the study included 282 adult transgender patients and 2370 age- and race-matched non-transgender controls. The mean age of the transgender cohort was 33 years, with 60% of these patients being White and 27% African American (Table 1).

Concerning healthcare utilization among transgender patients compared to controls, we observed substantially higher use of email/telephone visits, sign-on to the KP patient portal, increased cancellations and no-shows and increased Behavioral Health visits. Similar primary care utilization patterns were demonstrated between transgender patients and controls (Table 1). For transgender-specific characteristics, Table 2 shows that 77% of transgender patients were included in the registry based on at least two transgender-related diagnosis codes, and 70% had seen a preferred provider (preferred providers are physicians or mental health specialists, who undergo additional training related to transgender patient care). Only 6% of patients in the transgender registry had documentation in their EMR of a transgender procedure. While 56% of transgender patients had a physician order on record for hormone medication, we observed only 36% of the transgender patient population with a hormone fill. Out of 158 patients with hormone orders, 51 (32%) had no corresponding fill.

### 3.2. Qualitative Component: TGNC Patient Focus Groups Results

A total of 28 patients participated in one of four focus groups. These participants identified mostly as female (39%), were predominantly white (68%), and highly educated (74% with an associate’s, bachelor’s degree, or some college) (Table 3). Most of the focus group participants were on hormone replacement therapy (HRT) (89%); however, a minority had undergone a TGNC procedure (36%) (Table 3).

Analysis of the focus group discussions revealed six key themes (Table 4), including limited availability of TGNC information, negative and positive sentiments regarding providers, issues with case management, limited access to care, lack of coordination of care, and negative experiences with staff. The percent agreement across the 3 coders ranged from 49% to 59% for the key themes, with an overall agreement across coders for all key themes of 46% (Table 4). The Krippendorff’s Cu-α, another measure of inter-rater agreement that calculates agreement simultaneously for all selected themes, was 0.40. In addition, within each key theme, we explored the major co-occurring themes (Table 5), demonstrating how the themes intersect. The co-occurring themes further established areas of emphasis and aided in the prioritization of quote selection.

#### 3.2.1. Limited Availability of Information

As noted by many participants, one deficit in the quality of care was a lack of information about transgender care available to patients. Many participants actively sought information about transgender care from resources where it would be reasonable to expect information, such as the online patient portal called “kp.org”, or by calling member services. As a result of the lack of information, and the lack of coordination to direct patients towards appropriate resources, patients indicated that they were required to search for information independently through Internet searches or forums to learn from experiences of other transgender persons in their community. Information pertaining to benefits and the bureaucracy of the health system was an issue that generated many comments and frustration among the participants.

One of the critical reasons why it is urgent to address the lack of information available to transgender patients is the perception about why this gap exists. An unexpected view expressed by focus group participants was an assumption that health systems feared alienating cis-gender members by openly providing TGNC resources.


*[Regarding the online patient portal] “There’s not much there. There’s basically nothing.”*



*“Member services will tell you nothing.”*



*“How do you find that out [information on benefits]? Am I missing something? …It’s like you have to go on the Internet and find people saying something [about] Kaiser because it’s like nowhere. You look, and you look, and you look…”*



*[About transgender information] being more publicized. “I get why it’s not…I understand that, especially traditionally more conservative people who, if they hear that Kaiser is at the forefront of trans healthcare they might consider switching providers, which could be a problem. So, I kind of get why it’s kind of behind the curtain.”*


#### 3.2.2. Negative and Positive Experiences with Providers

Focus group participants expressed both positive and negative sentiments concerning their interactions with providers and confidence in their provider’s care and knowledge of transgender health. Participants seemed to feel that their doctors were adequately trained to treat medical conditions and health crises but were often not proficient in the specific health issues that affect transgender patients. Participants in one focus group unanimously agreed that a lack of knowledge on treating transgender patients is dangerous for transgender patients and results in negative treatment quality and potentially serious adverse health outcomes.

The lack of confidence that patients expressed regarding their physicians’ knowledge about transgender health extended beyond medical knowledge to include inappropriate patient–provider interactions. These interactions ranged from unseemly questions from provider to TGNC patient to feelings of anxiety, fear, and negative anticipation of appointments with their doctors. As a result of negative provider interactions, many patients indicated that they would avoid seeking healthcare unless it was necessary.

Conversely, there were numerous positive experiences described by focus group participants regarding interactions with their healthcare providers. The vast majority of positive sentiments were related to endocrinology and behavioral health. Participants expressed that their endocrinologists were comprehensive and knowledgeable about hormone therapy for transgender patients and included them, as patients, in decision-making issues related to the hormone therapy. They also described the thorough and caring nature of surgeons during the very stressful experience of undergoing transgender surgical procedures. Finally, there were several participants who said that they had overall good experiences with TGNC care, including interactions with physicians, and specifically the staff, who supported and coordinated care through the Transgender Health Services Program.


*“Sure. They understand hands and fingers and broken bones. They don’t understand transgender.”*



*“The first [doctor] … had absolutely no clue about transgender. And I say that because I know when I’m being talked to from a book, and that’s what he was doing. That was his experience level.”*



*“What happened to the basic tenet of ‘Doctor, do no harm?’…By ignoring, you’re doing harm. By not learning, you are doing harm. It’s by omission, you are doing harm.”*



*“I was going to a sleep doctor… and while he was working out the sleep study watch, he asked me if I had a penis or a vagina and I’m like, is that clinically relevant to getting my sleep cycle under control or is that a personal curiosity thing?”*



*“Personally, I don’t feel comfortable getting my physical done with my primary care doctor; simply because I—just a lot of the way that the questions are worded I’m just concerned.”*



*“When I go to the endocrinologist and I ask for injections and the endocrinologist looks at me like I just asked for a prescription for crack cocaine, I mean, that bothers me.”*



*“[My endocrinologist] was very knowledgeable and explained to me why she was making that decision and allowed me to be part of the discussion and so that if I had anything—if I saw anything in terms of side effects, I could let her know and we could look at what would be an alternative.”*



*“The endocrinologist was very knowledgeable about trans issue too. She’s had a lot of experience with hormone replacement therapy, so that was great too.”*



*“At the end of the consult [the surgeon] said, ‘Well, let’s make the next 50 years really good for you.’ So, he was really positive and very caring.”*



*“I have to say that once I got involved with a transgender team, then everything worked out a lot smoother and I’ve had great care from my therapist to my primary care physician and my endocrinologist to my urologist.”*


#### 3.2.3. Issues with Case Management

Participants voiced that one of the biggest healthcare issues was concerning case management. Case managers play a critical role for TGNC patients because they can serve as sources of detailed and specific information regarding transgender care. Issues included those related to accessing information, including benefits, and coordination of healthcare services and health providers, especially among patients who had undergone or were contemplating surgical procedures. Patients expressed the importance of having a case manager, but only if that person was knowledgeable, responsive, and sensitive to TGNC issues. Unfortunately, several participants were not aware that case managers were available to help them navigate the health system until the focus group discussions.

Discussions indicated that additional training and a standardized protocol are needed for transgender case managers to ensure consistency in the information provided. Further, staffing case managers who are dedicated and invested in their positions could make a difference for transgender patients, many of whom may already be stressed and anxious, while seeking transgender-specific care. Feedback from the focus groups indicated a need for knowledgeable, conscientious, and dependable case managers. Many individuals expressed that the case manager’s role was critical to obtaining quality healthcare for transgender patients because they are the gateway to essential information. Participants also remarked that a case manager who was transgender themselves would be more trustworthy and invested in their job and able to provide information and coordinate care in an understanding and compassionate manner.


*“I notice a lot of people talking about having like a case manager or advocate; I don’t have one. I didn’t know that that was a thing.”*



*“I had talked to my case manager multiple times and nothing happened. So, I just wanted to talk to somebody else.”*



*“I’m actually on my third case manager…. The first one left—well, I had been dealing with her for about two years while she said that it wouldn’t be covered, the surgery. And then I got another one, and she said that it would be covered, but she was only there for like another three weeks. She was leaving; she told me that. The first one never told me that she was leaving.”*



*“When I was talking with [the case manager] about one of the procedures that I wanted to get done, I had to explain over the phone what the procedure was. I was like, ‘Isn’t that your responsibility to look that up?’ But in my office at work, I had to explain it to her.”*



*“Yeah, I think you would want like your case manager to be like the strongest link in the whole thing and yet it seems like it’s sometimes like the weakest link.”*



*“I think the case manager, the trans specific case manager thing, is huge because without that it we would be even more [in the dark].”*



*“I think you need a transgender person in the position [of case manager] who has some experience to be able to talk with transgender folk or at least if you can’t find someone transgender, someone who’s willing to coordinate with someone and get some kind of protocol in place for that person.”*


#### 3.2.4. Limited Access to Care

In every focus group, issues related to access to healthcare services were discussed. Several participants were vocal about difficulties they had concerning hormone therapy, including the cost and availability of prescribed medications and the availability of providers that were located close to their home or home medical facility. Access to care was mentioned frequently with ‘availability of TGNC information’ (Theme 1). The lack of information was perceived as the health system hiding benefits and services that TGNC patients need.


*“Nobody in endocrinology felt comfortable providing that care for a transgender person. So, they recommended an outside provider. I sought it out and found an outside provider, and I was denied because it was considered to be sex change related, although I had been on this hormone therapy for fifteen years at that point.”*



*“I go to Whitman & Walker in D.C. [because] it ends up being cheaper. My entire year’s supply of hormones—I worked it out, it’s about $200. And that’s paying like full price out of pocket …. But even then, like tests and medication and everything, it’s maybe $500 a year [which is cheaper than] what the quoted Kaiser prices were like when I asked.”*



*“My primary care is Fredericksburg but the nearest therapist that specializes in transitions is in Burke, and the nearest endocrinologist is Falls Church. I just don’t want to get like pushed to where I have to drive an hour for each appointment.”*



*“[Care] is not accessible because they’re hiding it; like Kaiser Permanente is doing it in the background you know. I get thatfrom a perspective of like an organization you may have some kind of like negative comment from other populations. But I don’t think that should be a reason for the organization to be not as visible.”*



*“People shouldn’t be finding out by accident about a case manager. You should know what your benefits are. And to be told by the company that’s going to provide them or not provide them two completely different stories is wrong. It puts you on pins and needles, and the process is hard enough without that.”*


#### 3.2.5. Lack of Coordination of Care

Coordination of care was frequently described as an issue because of the number of different providers patients must consult for issues related to their TGNC status. Patients who decided to undergo transition surgeries had additional serious points of coordination that were problematic. The main co-occurring theme with coordination of care involved case management. Patients felt like the case manager should serve as a point person for their care, which was not what they experienced. Coordination of care was also frequently brought up concerning hormone treatment and/or care with an endocrinology provider, and many patients discussed issues they had maintaining their existing treatment regimen when they first enrolled in the health system.


*“I didn’t get like a step-by-step plan at any point. They’re just like, “Oh, first, you need to go see this guy.” So, then I go see him and he writes the letter, and he says, “Okay, talk to your case manager,” and then I talk to my case manager and she’s like, “No, he’s supposed to refer you to the next person.” So, then I go back to him and he refers me to the second person, and then the same thing happens.”*



*“[You need a case manager] who’s willing to coordinate with the patient and get some kind of protocol in place for that person because I feel like they’re just running around in circles because they have no idea what to do.”*



*“Two days prior to my scheduled surgery …my cardiologist, my surgeon and my primary care physician were not communicating… I had to [find out] what medication I should be taking, which I shouldn’t be taking a few days before the surgery. So, I still kept taking because I didn’t get any notification that I should stop.”*



*“When I changed from another [health system] I just wanted someone to continue my hormone regimen and the assigned primary care physician wasn’t comfortable, so he wanted someone from endocrinology to do it. Nobody in endocrinology felt that they were comfortable providing that care for a transgender person, so they recommended an outside provider.”*



*“It took me a span of probably about six months [to get hormone treatment]. I called my primary care, who then called my psychiatrist, who then called me to say, “Hey, you need to schedule something with this person up in Burke. And then I go back up there, and he was like ‘I have no idea why they didn’t put you on hormone.’”*


#### 3.2.6. Negative Experiences with Staff

The majority of negative staff experiences were related to interactions with case managers. While most of the issues regarding case managers were concerning knowledge, responsiveness, and abilities, there were also common sentiments about the personal communications being uncomfortable, such as having to remind the case manager of the correct pronouns. Most of the discussion around misgendering and inappropriate communication came from interactions with clinical and administrative staff, pharmacists/pharmacy technicians, and even providers. The description of interactions ran the gamut of accidental misgendering, to staff purposefully behaving in a hostile manner and refusing to acknowledge a patient’s TGNC status, and often occurred in an open area of a medical center. As noted in the focus groups, these negative interactions are impactful because it often prevents TGNC patients from seeking needed care, and it places an additional emotional strain on individuals, who may already be struggling with mental health issues. Because of these negative interactions, several patients recommended the need for specialized staff training.


*“When it comes to pronouns and receptionists… I get misgendered because of perceptions from receptionists, but there’s nothing on the file that says anything other than my biological sex. So, they just kind of go with that.”*



*“And then I went to pick up [hormones] and the pharmacist says to me, ‘What are you going to use that for? For a horse?’”*



*“I had to argue with the medical assistant about what my name is and what my gender was. So, I just walked out of the appointment. Apparently, the medical assistants are not told to look at the chart before you see a patient.”*



*“I’ve had to explain to people on the phone [that I’m transgender] and sometimes they still don’t get it. And that’s just an uncomfortable thing. I think once after I explained it and completely outed myself, I would hope they would know what to do and run with it, but I still got “sir-ed” at the end of the conversation.”*


## 4. Discussion

Our cross-sectional descriptive analysis of 282 transgender patients demonstrated that primary care and emergency care utilization were similar for both transgender and non-transgender insured patients. However, transgender patients had higher numbers of visit cancellations or no-shows, more email and telephone encounters, and higher utilization of the online patient portal than non-transgender controls. These results are consistent with reports in the literature of anxiety and fear of discrimination that transgender persons experience in the healthcare setting leading to avoidance or delays in seeking care. These sentiments of fear and anxiety and delaying needed care were also echoed in the patient focus groups. In terms of pharmacy utilization, we found that among transgender patients, who have prescription orders for HRT, 32% are not filling their medication at KP. Reasons for this finding were signaled by focus group participants, who described HRT to be too expensive or unable to obtain the type of HRT that they desired within the health system.

The focus group discussions (among 28 patients of diverse gender identities) revealed several distinct themes about the quality of care from the patient perspective, including positive and negative sentiments regarding patient–provider interactions, access to and availability of transgender information, and shared negative experiences with their case manager. While some of these findings are supported by previous research, some findings are unique. However, all should be taken into consideration as part of an improvement plan for TGNC health programs.

Research on gaps in healthcare for transgender patients has been previously reported [5,6,7,8], including one of the largest studies, the National Transgender Discrimination Survey (NTDS) [7], which was a 70-item paper and online questionnaire with 6456 respondents nationwide. This survey indicated that 28% of participants reported verbal harassment within the medical setting, and 2% reported being physically attacked in the doctor’s office when seeking medical care. Not surprisingly, 21% of the NTDS respondents reported that nobody in their healthcare system knew that they were transgender. In fact, providers’ awareness of their patient’s transgender status increased experiences of discrimination by up to 8% among participants depending on the setting, including being denied service altogether. Among respondents whose providers knew they were transgender, 29% reported that they delayed seeking care when ill; and 33% postponed preventive care because of discrimination by providers.

Transgender patients (not only in our study but in numerous other studies) [2,6,9,10] report that their greatest barrier to optimal transgender healthcare is the lack of knowledgeable providers. The lack of knowledge and training of providers regarding the health needs of transgender individuals puts transgender patients at risk of medical errors and inappropriate treatment. Sanchez et al. conducted interviews with 101 male-to-female transgender persons from 3 community medical centers in New York City to assess barriers to care [9]. The most frequent barrier to care reported by 32% of the interview participants was access to a provider who was knowledgeable about transgender healthcare. The lack of transgender medical education also has been well-documented. While most medical schools report teaching gender identity, the quality of the coverage of LGBT-related topics has been rated predominantly poor, very poor, or fair (70%). Only one-third of medical schools reported having any curriculum related to hormonal and surgical transitioning [11].

The relationships and interactions that transgender patients have with their providers are crucial. Fear of stigma and discrimination in the healthcare setting prevents patients from accessing care and disclosing their gender identity [6,12]. In a 2014 publication of a secondary analysis of the National Transgender Discrimination Survey [6], a study examined reasons and differences for postponement of primary curative care in a sample of 4049 respondents. It demonstrated that discrimination was the major predictor associated with postponement of care and that this association differed based on natal sex. Further, respondents who were “out” (or open about their gender identity) to their provider were nearly twice as likely to postpone care due to discrimination than respondents who were not “out.”

There is a fundamental issue related to the availability of information regarding transgender health issues [13,14,15]. Two types of information-seeking challenges were identified by focus group participants: (1) broad information on transgender health and medical and social transitions; and (2) specific information on navigating the health system. Within the health system of this study, there was no systematic dissemination of education and information resources that transgender patients were aware of. The most obvious places that patients would expect to find information, such as the online patient portal and member services, should be the starting point for providing, at minimum, basic information/resources about transgender healthcare services. Without resources within the health system, patients seek out information in the community and on the Internet.

An additional issue is the accessibility and quality of the information available to patients on transgender health and health care issues. For example, a study by Vargas et al. conducted a systematic search of online information resources for gender-affirming surgery and discovered that identifying patient-directed information required excessive navigation through websites of non-clinical resources [15]. This is different from searches on other common and similar medical topics, such as “breast cancer surgery” or “hernia surgery”, which produces results from more well-known health websites. Further, none of the websites that were identified by this study met the recommended standard of a 6th-grade reading level for health-related materials. This excessive amount of information and lack of appropriate literacy levels pose significant barriers to patients looking for reliable resources on transgender healthcare.

Case management issues were consistently voiced by TGNC patients, who participated in the focus groups. In many ways, it seems like improving case management for transgender patients should be the priority for improving healthcare quality. As noted by others [16,17], case managers who work with transgender patients require knowledge of unique concepts, communication skills, and health needs for which they may have had no training. For instance, the case manager may be responsible for being an advocate and educator, finding culturally competent care, verifying and explaining benefit coverage and network requirements for services, screening and addressing behavioral health issues, and addressing social determinants of health. In a recent study of EMR data conducted at Fenway Health [18], case management was identified as a system-level factor that was associated with the utilization of outpatient behavioral health services. This finding suggests improved case management for TGNC patients could also facilitate engagement with behavioral medicine when needed as mental health issues have been shown to be common in the transgender and nonbinary patient populations [19,20,21].

Finally, an indirect finding from our focus group analysis was the need for peer-to-peer support among TGNC patients. Many participants noted that the focus group was the first time they had spoken to another TGNC patient within the same health system, and some further noted that the discussion marked the first time they had spoken to any fellow TGNC individual. The focus groups were originally planned for 2 h in duration. However, in every focus group, additional time was required to allow participants to continue conversations, share additional experiences, and ask other participants questions. Recent research has underscored the importance of social support for transgender individuals, specifically linking social support to lower levels of depression and anxiety, suicidal behaviors, and improved quality of life [22,23,24].

The main limitations of the quantitative component of the study were the small sample size and lack of sexual orientation and gender identity data available at the time of the analysis. In addition, using diagnostic codes to identify TGNC patients may not have captured all TGNC patients resulting in a non-representative sample. However, leveraging electronic medical record data to provide a preliminary description of the transgender population was meaningful and provided baseline data with which to evaluate measures over time. For the qualitative component of the study, small sample size was also an issue, as was the low percent agreement between coders. Because this was a pilot study, we were restricted to a smaller number of participants, and the study staff was not able to go back to discuss coding issues to attain a higher inter-coder agreement. However, the issues documented in this paper and the corresponding recommendations were consistent with results from a qualitative analysis among providers at the same health system regarding the quality of care for TGNC patients [25]. Further, like many other studies among TGNC persons, our sample for the qualitative analysis was predominantly white (due to convenience sampling). Future studies should consider targeted and oversampling approaches to recruiting adequate numbers of racial/ethnic minority populations with input from a diverse patient stakeholder group.

## 5. Conclusions

Our study describes a small but growing insured TGNC patient population at a healthcare system in the Mid-Atlantic region. We documented issues related to the quality of TGNC healthcare according to 28 patients, who participated in focus groups and concluded four key recommendations for health systems to improve the quality of care of TGNC patients.

First, healthcare systems should develop and maintain a TGNC website with educational material, information on preferred providers, FAQs, and checklists on navigating the health system. Second, consistent and sustained cultural competency training should be provided to primary care and specialty physicians on TGNC patient interactions and specific TGNC health issues. Consistent and sustained cultural competency training should also be provided to administrative and clinical staff on TGNC patient interactions and specific TGNC health issues. Third, healthcare systems should increase the availability of digital care, including telemedicine, to address healthcare access issues and reduce patient anxieties and fear of discrimination in the healthcare setting. Lastly, we recommend hiring a team of TGNC-specific case managers who are trained and knowledgeable about the TGNC patient population. This team should be sufficient in numbers to effectively provide patient support and coordinate care for the growing number of TGNC patients.

## Figures and Tables

**Table 1 healthcare-09-00530-t001:** Characteristics of transgender patients age- and race-matched controls.

	Transgender Cohort *N* = 282	Age- and Race-Matched Controls *N* = 2370
**Demographics**		
Age, mean (SD)	32.6 (13.9)	33.3 (13.7)
Male *, *n* (%)	122 (43)	1325 (56)
Female *, *n* (%)	160 (57)	1045 (44)
**Race/Ethnicity, *n* (%)**		
American Indian/Alaskan Native	1 (0)	10 (0)
Asian/Pacific Islander	9 (4.0)	90 (4.0)
Non-Hispanic Black	63 (27)	630 (27)
Hispanic	24 (10)	240 (10)
Multiracial	1 (0)	10 (0)
Non-Hispanic White	139 (59)	1390 (59)
Missing, *n*	45	0
**Insurance type, *n* (%)**		
Commercial	210 (79.9)	2042 (86.2)
Medicaid	36 (13.7)	217 (9.2)
Medicare	17 (6.5)	111 (4.7)
Missing	17	0
**Healthcare utilization **, mean (SD)**		
Outpatient visits	19.22 (17.0)	10.95 (11.0)
Urgent care visits	0.91 (1.6)	0.91 (1.7)
Ed visits	0.46 (1.3)	0.26 (0.9)
Inpatient stays	0.28 (1.1)	0.15 (1.0)
Email/telephone encounters	20.62 (18.6)	8.94 (13.2)
Behavioral health encounters	9.75 (14.7)	1.37 (5.4)
Outpatient visits in primary care	4.12 (4.8)	3.72 (4.2)
Completed encounters	19.5 (20.8)	10.19 (11.8)
Canceled encounters	6.76 (9.0)	3.62 (6.8)
No-show encounters	3.45 (5.9)	2.14 (3.6)
Patients with >3 PCPs ^†^	19 (0.07)	353 (0.15)
Patient portal sign-ons per person 2015	40.05 (37.4)	21.29 (22.8)
Patient portal sign-ons per person 2016	43.81 (36.2)	22.76 (24.7)

ED = emergency department; PCP = primary care provider; * gender reported in the electronic medical record; ** healthcare utilization was measured during the 24 month lookback period; ^†^ Patients with outpatient visits with more than three different PCPs.

**Table 2 healthcare-09-00530-t002:** Characteristics specific to the transgender cohort.

	%
**Transgender diagnoses, *n* (%)**	
Minimum of 1 transgender diagnosis	87
Minimum of 2 transgender diagnoses	77
Minimum of 1 transgender diagnosis from BH/END	71
**Patient has seen preferred provider, *n* (%)**	70
**Transgender procedures, *n* (%)**	
Patients with procedures indicating MTF	4
Patients with procedures indicating FTM	2
**Hormone medication orders, *n* (%)**	
Patients with medication orders indicating FTM	21
Patients with medication orders indicating MTF	35
**Hormone medication fills, *n* (%)**	
Patients with medications indicating MTF	23
Patients with medications indicating FTM	13
**Patients with hormone medication order and no fill (*n* = 158)**	32

BH = Behavioral Health; END = Endocrinology; MTF = male-to-female; FTM = female-to-male.

**Table 3 healthcare-09-00530-t003:** Description of transgender patient focus group participants (*N* = 28).

Characteristic	%
**Gender Identity**	
Female	39
Male	11
Non-binary/genderqueer	18
Not sure/questioning/other	8
Transman	21
Transwoman	4
**Race**	
White	68
Other	32
**Education**	
High school or equivalent	7
Associates, Bachelors, some college	74
Masters, doctoral, or professional degree	19
**Current Hormone Use**	89
**History of any transgender procedure**	36

**Table 4 healthcare-09-00530-t004:** Predominant transgender patient focus group themes.

	Frequency of Occurrence	Percent Agreement across Coders *
Limited availability of TGNC information	207	53
Negative experience with provider	133	56
Negative experiences with case management	132	49
Limited access to care	128	54
Coordination of care	88	56
Positive experience with provider	76	56
Negative experiences with staff	72	59

TGNC = transgender and gender nonconforming; * percent agreement across all predominant themes considered together was 46%

**Table 5 healthcare-09-00530-t005:** Transgender patient focus group coded themes.

Predominant Themes (Frequency)	Primary Co-Occurring Themes/Codes
Limited availability of TGNC information (207)	Case management
	Coordination
	Access to care
	Surgery
	Bureaucracy
Negative experiences with provider (133)	Primary care
	Availability of information
	Negative experiences with staff
	Endocrinology
	Case management
	Access to care
Negative experience with case management (132)	Availability of information
	Coordination
	Negative experiences with staff
	Negative experiences with provider
	Access to care
	Surgery
Limited access to care (128)	Endocrinology
	Availability of information
	Behavioral Health
	Bureaucracy
	Case management
	Coordination
Coordination of care (88)	Case management
	Availability of information
	Endocrinology
	Access to care
	Negative experiences with provider
Positive experience with provider (76)	Primary care
	Endocrinology
	Quality of care
	Positive experience with staff
Negative experiences with staff (72)	Case management
	Negative experience with provider
	Availability of information
	Coordination of care
	Primary care

TGNC = Transgender and Gender Nonconforming.

## Data Availability

The data used for this study contain protected health information (PHI). Therefore, the data are available upon request to researchers who meet the criteria for access to confidential data with approval from Kaiser Permanente Mid-Atlantic States (KPMAS) and the KPMAS IRB.

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
