# Peer review of "A Mixed Methods Study Describing the Quality of Healthcare Received by Transgender and Gender Nonconforming Patients at a Large Integrated Health System"

_healthcare, 2021, doi:10.3390/healthcare9050530_

Round 1

Reviewer 1 Report

The authors present a mixed-methods analysis of the care of transgender-identified patients of Kaiser Permanente Mid-Atlantic States. The analysis is thoughtful , well-grounded, and insightful.

A quick note about language - I'm sure the authors are well aware that these are in no sense a "random" population of transgender people, and they do not pretend they are. But perhaps terms like "patients seeking gender-affirming care" or some such would be a more accurate description of their quantitative and qualitative populations.

A few minor points to consider:

1) Was any procedure applied to confirm the gender identity of persons identified as "transgender" on the basis of ICD codes? In other samples (I believe from KP, if I'm not mistaken), these codes sometimes identify patients who are discussing the transition of children or romantic partners, as opposed to an accurate reflection of self-identification. It shouldn't be too hard to to a medical record search on only 282 patients to sort that out.

2) For the quantitative sample, describe the matching procedure a little more clearly - for instance was the age matching within 1 year? Why were patients seeking gender-affirming care with no identified race/ethnicity not matched? These patients should either be matched in the same way (to comparators with no identified race/ethnicity), or dropped from the analysis for the sake of comparability. Also, specify that there were 10 comparator patients selected per patient seeking gender-affirming care.

3) In table 2, you don't need both n's and %'s. Also, these are not percents, but proportions (e.g. 0.77 is 77%).

Reviewer 2 Report

This is a great paper, very few revisions suggested:

Title- could be more refined to highlight what aspects of healthcare and where the study is situated. Sounds quite generic at the moment.

Abstract- where did the study take place? Typically written in past tense. 

Methods- 2-3 hour focus groups are exceptionally long, but also really positive because it indicates that people felt comfortable talking. Can you expand on the parameters of this part of the methods a little to provide insights that may be of use to other researchers/community groups? 

Sample- like most transgender studies, the sample is primarily white. Any insights or ideas for how to reach further into other racialized populations to get a more nuanced, representative set of experiences? Race isn't mentioned in the discussion of the study limitations, but it certainly could be. 
